# Design, Manufacturing, Characterization and Evaluation of Lipid Nanocapsules to Enhance the Biopharmaceutical Properties of Efavirenz

**DOI:** 10.3390/pharmaceutics14071318

**Published:** 2022-06-21

**Authors:** Grady K. Mukubwa, Justin B. Safari, Roderick B. Walker, Rui W. M. Krause

**Affiliations:** 1Department of Chemistry, Faculty of Science, Rhodes University, P.O. Box 94, Grahamstown 6140, Eastern Cape, South Africa; gradymukubwa@gmail.com (G.K.M.); safaribazibuhejustin@gmail.com (J.B.S.); 2Department of Pharmaceutics, Faculty of Pharmaceutical Sciences, University of Kinshasa, Kinshasa XI B.P. 212, Democratic Republic of the Congo; 3Department of Pharmacy, Faculty of Pharmaceutical Sciences and Public Health, Official University of Bukavu, Bukavu B.P. 570, Democratic Republic of the Congo; 4Department of Pharmaceutics, Faculty of Pharmacy, Rhodes University, P.O. Box 94, Grahamstown 6140, Eastern Cape, South Africa; 5Center for Chemico- and Biomedical Research, Rhodes University, P.O. Box 94, Grahamstown 6140, Eastern Cape, South Africa

**Keywords:** viral resistance, lipid nanocapsules, efavirenz, design expert

## Abstract

Despite their incredible contribution to fighting viral infections, antiviral viral resistance is an increasing concern and often arises due to unfavorable physicochemical and biopharmaceutical properties. To address this kind of issue, lipid nanocapsules (LNC) are developed in this study, using efavirenz (EFV) as a drug model. EFV solubility was assessed in water, Labrafac Lipophile and medium chain triglycerides oil (MCT oil). EFV turned out to be more soluble in the two latter dissolving media (solubility > 250 mg/mL); hence, given its affordability, MCT oil was used for LNC formulation. LNC were prepared using a low-energy method named phase inversion, and following a design of experiments process. This one resulted in polynomial models that predicted LNC particle size, polydispersity index and zeta potential that were, respectively, around 50 nm, below 0.2 and below −33 mV, for the optimized formulations. Once synthesized, we were able to achieve an encapsulation efficacy of 87%. On the other hand, high EFV release from the LNC carrier was obtained in neutral medium as compared to acid milieu (pH 4) with, respectively, 42 and 27% EFV release within 74 h. Other characterization techniques were applied and further supported the successful encapsulation of EFV in LNCs in an amorphous form. Stability studies revealed that the developed LNC were quite stable over the period of 28 days. Ultimately, LNCs have been demonstrated to improve the biopharmaceutical properties of EFV and could therefore be used to fight against antiviral resistance.

## 1. Introduction

For several decades, humanity has been confronted with various types of infectious diseases, including viral diseases, such as the recent coronavirus pandemic that has claimed millions of lives. Many strategies have already been studied and implemented in the battle against viral infections, including natural medicines, chemical drugs and biotechnology-derived medicines as well as vaccines [1,2,3]. Among these approaches, chemical antiviral drugs are the most used as a primary resource to fight viral infections [4]. Unlike antibacterials, antiviral drugs are very difficult to design owing that viruses use the host’s biosynthetic functions to replicate and are prone to many mutations [5].

Researchers have investigated and developed antiviral drugs that safely use limited human biosynthetic functions as targets. Most of these targets are specific for each virus, so that broad-spectrum antiviral drugs are difficult to develop. Although thousands of antiviral drugs have already been reported in the literature, only around 90 of them are formally approved to date for use in humans [5]. This makes them very precious compounds that must be well used and protected for their effectiveness and prolonged use against viral diseases.

Despite their famous success in viral infections therapy, antiviral drugs are likely to face resistance and become ineffective. In actuality, many factors can negatively affect antiviral drugs efficacy encompassing their physicochemical properties together with their metabolism. Thus, a thorough understanding of their structural, physicochemical, kinetic and bioactive characteristics is very useful to improve antiviral drugs efficacy [6,7,8].

Antiviral drug therapy is mainly hindered by the solubility and permeability behavior of the active ingredients [4]. According to the biopharmaceutical classification system (BSC) whose elaboration is based on the dose number (*D*_0_) and the oil–water partition coefficient (log P), *D*_0_ values superior to one and log P values inferior or equal to 1.632 are, respectively, considered as the low solubility and low permeability criteria. Based on these two parameters, the BSC splits up antiviral drugs into three groups. Fifty percent of them exhibit “low solubility” and belong to the BSC II class (e.g., arbidole, peramivir, letermovir, remdesivir, ritonavir, baloxavir, chloroquine, marboxil and lopinavir), while 44% have “poor permeability” and belong to the BSC III class (e.g., acyclovir, lamividune, ribarvirin, oseltamivir, entecavir, zidovudine and adefovir). Those antiviral drugs that exhibit both low solubility and low permeability (BSC IV) account for 6% (e.g., favipiravir) [4]. This illustrates how critical these two parameters are in the fate of antiviral drugs in the body, especially their absorption in vivo.

Researchers have investigated several strategies to overcome these drawbacks, and, among them, loading antiviral drugs into advanced delivery systems, such as nanomaterial-based systems, have demonstrated promising outcomes [6,7,8,9]. In particular, lipid-based delivery systems have been successful, especially for drugs belonging to the BSC classes II, III and IV. There are many types of lipid-based delivery systems, including liposomes, microemulsions, nano-emulsions and nanostructured lipid carriers [9], and some antiviral drugs have successfully reached the market as lipid-based formulations. This is the case for amprenavir (Agenerase^®^), saquinavir (Fortovase^®^) and ritonavir (Norvir^®^) [10].

In this work, we aim to develop lipid nanocapsules (LNCs) to improve the biopharmaceutical properties of efavirenz (EFV), a non-nucleoside reverse-transcriptase inhibitor (NNRTI), with very poor solubility (BSC II) used, in combination, for first line treatment of HIV and recently investigated for its potential activities against SARS-CoV-2 [11,12]. LNCs are lipid-based drug-delivery systems introduced by Hertlault et al., in 2002, as stable systems with particle sizes ranging from 20 to 100 nanometers, depending on their components and proportions. They are easy to produce by use of an organic solvent-free low-energy method known as phase inversion [13,14]. Their structure, unlike more commonly used liposomes, are made of an oily liquid core surrounded by a rigid shell, suggesting their suitability for efavirenz owing to its log P of 4.6 [15,16,17]. LNCs exhibit the high-encapsulation efficiency of hydrophobic or amphiphilic drugs and possess a remarkable stability that could stand up to 18 months. They have been used for the encapsulation of a large variety of drugs, including anti-infectious agents, anticancer agents and active ingredients used for the central nervous system [18,19]. LNCs are considered to be promising vectors for the delivery of these therapeutic molecules by different routes, including oral, intravenous, cutaneous, cerebral and pulmonary [20].

On the other hand, Labrafac^®^ Lipophile 1349, Lipoid^®^ S75-3 and Solutol^®^ HS15 are the main constituents used to manufacture LNCs. To improve the affordability of these systems, we attempt to use different materials, namely, medium-chain triglycerides from coconut and palm kernel oil, crude soy lecithin and Tween 80. Since the critical quality attributes of LNCs include droplet size (DS), polydispersity index (PS) and zeta potential (ZP), design and optimization studies are implemented using the response surface methodology. This one was developed to study the relationship between different independent variables and one or more dependent variables. The response surface methodology mainly aims at generating information in the experimental domain of interest, while providing a reliable estimation of the experimental variability known as pure error. It also allows for the prediction of the observed response as closely as feasible in areas of the experimental domain in which no experiments have been conducted, and guarantees that the experimental data fit the suggested models [21,22]. I-optimal mixture design, the randomized optimal design and the “factorial multilevel categoric” design were selected, respectively, to ascertain the interactive effects of components on LNC attributes to evaluate the encapsulation efficiency and drug loading capacity of the developed nanomaterials, and to determine the drug release profiles.

## 2. Materials and Methods

### 2.1. Material

#### 2.1.1. Chemical

Crude soy lecithin granules were purchased from Health Connection Wholefoods (Cape Town, South Africa). According to the manufacturer’s specifications, these granules mainly contained phosphatidylcholine, phosphatidylinositol, polyunsaturated fat, saturated fat, glycemic carbohydrates and sodium. MCT oil was obtained from Absolute Organix (Johannesburg, South Africa), Labrafac Lipophile 1349 was purchased from Gattfossé (Saint-Priest, France), Tween 80 was purchased from Merck (Johannesburg, South Africa) and efavirenz was donated by Adcock Ingram Limited (Wadeville, South Africa). Sodium chloride was purchased from Minema (Roodepoort, South Africa). HPLC-grade acetonitrile was purchased from Merk (Darmstadt, Germany). HPLC-grade water was produced using a RephiLe Bioscience Direct-Purefi Ultrapure RO Water system (Boston, MA, USA).

#### 2.1.2. Instrument

A sonication bath (Digital Ultrasonic Cleaner PS-10A, Meizhou, China) was used during the development process. A Benchmark “my Fuge” mini centrifuge (Benchmark Scientific, Sayreville, NJ, USA). HPLC analysis was performed using an Agilent 1100 Liquid Chromatography series equipped with a quaternary pump (G1311A), degasser (G1322A), diode array detector (G1315B) and manual injector (G1328B) with a Phenomenex^®^ Kinetex^®^ column (2.6 μm C18, 100 Å, 150 × 4.6 mm i.d.). Samples were freeze-dried using a LABCONCO FreeZone^®^ 6 Liter Benchtop Freeze-Dry system (Kansas City, MO, USA). A PerkinElmer Spectrum 100 FT-IR Spectrometer was used to record IR spectra and a TA DSC 250 instrument was used for thermal analysis. The material crystallinity was assessed using an XRD D8 Discover or D2 Phaser Instrument (Bruker, Billerica, MA, USA). A Zetasizer nano ZEN–3600 MAL1043132 from Malvern Instruments (Malvern, UK) was used to determine the particle size, polydispersity index and zeta potential of the materials. Particles’ shape was analyzed using a Zeiss Libra-120KV TEM instrument (Oberkochen, Germany). The elemental composition of the developed materials was evaluated by using an INCA PENTA FET coupled to VAGA TESCAM energy-dispersive X-ray spectroscopy (Brno, Czech Republic).

### 2.2. Methods

#### 2.2.1. Quantitative Determination of Efavirenz

A reversed-phase high-performance liquid chromatographic method developed by Bienvenu et al. was validated for the quantification of EFV [23]. The validated method was applied in the solubility, encapsulation and drug loading capacity, and in vitro drug release studies. The mobile phase for the reversed-phase isocratic elution consisted of a 75:25 *v/v* mixture of acetonitrile and HPLC-grade water with a pH adjustment to 3.2 using 0.1% formic acid. The mobile-phase flow rate, analysis time and injection volume were, respectively, 1 mL/min, 3 min and 20 µL with the detection wavelength fixed at 247 nm for EFV monitoring. The method was linear over the concentration range of 5 to 125 µg with R^2^ = 0.9995 and precise with %RSD < 4% for all the tested samples.

#### 2.2.2. Solubility Assessment of Efavirenz

Efavirenz solubility was evaluated by adding an excess amount (1.5 g) to 5 mL of either Labrafac Lipophile 1349, MCT oil or HPLC-grade water. Then, shaking was performed for 24 h at 750 revolutions per minute (rpm) at 25 °C, followed by centrifugation at a speed of 6000 rpm for 30 min. The supernatant was thereafter treated with acetonitrile–water (40:60) to precipitate the lipids. Subsequently, it was filtered using a 0.22 μm simplepure™ syringe filter, and the efavirenz content in the filtrate was determined using the validated HPLC method described in Section 2.2.1.

#### 2.2.3. Design and Optimization of Lipid Nanocapsules

##### I-Optimal Mixture Design and Statistical Optimization

With the aid of Design-Expert software version 13.0 (Stat-Ease, Inc., Minneapolis, MN, USA), the I-optimal mixture design was performed to ascertain the interactive effects of different proportions of LNC components on the droplet size, polydispersity index and zeta potential. This design was selected given that it provides lower average prediction variance across the region of experimentation and is desirable for response surface methodology for which prediction is important [24]. The MCT oil, crude soy lecithin, Tween 80 and NaCl-water were the independent variables while the droplet size, polydispersity index, zeta potential and temperature of dilution were the responses. These responses were expected to fit the different polynomial models for prediction purposes. The independent variables aligned with the ranges listed in Table 1 with the components’ proportions totaling 100%.

##### Model Optimization and Confirmation

The goal of this section was to assess the prediction performance of the polynomial models fitted to the responses. For this purpose, four different blends generated by the design expert software were selected and run in triplicate. The average values of the experimental data were compared to a 95% prediction interval (95%PI).
95%PI=^y0±t∝2, residual df×SEpred
where ^y0 is the predicted value of the response, $t_{\left(\frac{\propto }{2},\;residual\;df\right)}$ is the student’s *t* critical value and $SE_{pred}$ is the standard error of the prediction.

As presented in Table 2, the optimization criteria consists of maximized or minimized proportions of Tween 80 or MCT oil with proportions of crude soy lecithin and NaCl-water maintained in the fixed ranges of 1.5–3% and 71–79.5%, respectively.

#### 2.2.4. Preparation of Lipid Nanocapsules

The I-optimal mixture design from the Design Expert software version 13.0 generated 24 and 4 blends, respectively, for the design and optimization studies of LNCs. These were prepared using an organic solvent-free phase-inversion method. Briefly, as inspired from Heurtlault et al., a total weight of 2.5 g of all components (MCT oil, crude soy lecithin, Tween 80 and NaCl-water with NaCl amount fixed at 50 mg in all formulations) were mixed together in a round boiling flask under 1300 rpm magnetic stirring in an oil bath mounted on a hot plate. The temperature was raised from room temperature to 95 °C and three temperature cycles were applied in a range of 70 to 95 °C where phase inversion resulting in translucent mixtures was observed. Then, to obtain stable LNCs, a sudden cooling of the translucent mixtures with cold HPLC-grade water (1 to 4 °C), three times the volume of NaCl-water, was applied during the last cycle. A condenser was always set up to avoid any evaporation.

#### 2.2.5. Characterization

##### Droplet Size, Polydispersity Index and Zeta Potential

The Malvern Nano-ZS Zetasizer (Malvern Instruments, Worchester, UK) was alternately set to dynamic light-scattering mode for the droplet size (DS) and the polydispersity index (PDI) determination, and to laser Doppler anemometry (LDA) mode for the zeta potential (ZP). A dilution of 500 µL of the saturated samples in 50 mL of HPLC-grade water was performed prior to all determinations. The sample was placed into a 12.5 × 12.5 × 45 mm BRAND^®^ disposable cuvette (BRAND GmbH + CO KG, Wertheim, Germany) for the DS and the PDI, while a folded capillary cell was used for the ZP, and the experiments were performed at 22 °C at a scattering angle of 173°. The same parameters were evaluated over a period of 28 days for stability studies using samples stored at 15 °C + 0.8.

Subsequently, a drop of each saturated sample was placed onto a 3.05 mm holey carbon copper grid (FORMVAR/Carbon support 300 mesh) obtained from TAAB Laboratories Equipment, Ltd., Alderson, Berks, UK, for microscopic visualization of the LNC. The sample was stained with uranyl acetate for better visualization. A Whatmanfi 110 hydrophilic filter paper (Whatmanfi International, Ltd., Maidstone, UK) was used to adsorb excess liquid and the sample was allowed to dry for 24 h at room temperature (25 °C). Then, transmission electron microscopy (TEM) (accelerating voltage of 80 kV) was used to observe the shape of the LNCs. TEM images were processed using ImageJ software version 1.53e to allow for the establishment of the size distribution.

##### Encapsulation Efficacy and Drug Loading Capacity

The D-optimal (custom) randomized design from Design-Expert software version 13.0 (Stat-Ease, Inc., Minneapolis, MN, USA) generated 13 formulation runs for the optimization of the encapsulation efficiency and the drug loading capacity. Briefly, efavirenz, the independent input in this study, was added to an optimized LNC formulation in a range of 95 to 250 mg. After each preparation, a 1:4 dilution was performed followed by centrifugation at 6000 rpm for 30 min. The supernatant was treated with acetonitrile–water (40:60) and filtered using a 0.22 μm simplepure™ syringe filter, and the efavirenz content in the filtrate was determined using the validated HPLC method previously described in Section 2.2.1.

The following formulas were used to determine the encapsulation efficiency and drug loading capacity:(1)$$\%EE=\frac{Mf}{Mi}\times 100$$
where %*EE* stands for encapsulation efficiency, *Mf* is the efavirenz mass determined from the filtrate and *Mi* the initial mass of efavirenz in the formulation.
(2)$$\%DLC=\frac{Mf}{Mt}\times 100$$
where %*DLC* stands for drug loading capacity, *Mf* is the efavirenz mass determined from the filtrate and *Mt* the total mass of the formulation.

##### Differential Scanning Calorimetry

Differential scanning calorimetry allowed for the thermal assessment of the developed EFV-LNC as compared to that of the blank-LNC, the physical mixture as well as that of the raw materials. Briefly, the LNC dispersions were freeze-dried overnight. Then, 3 to 5 mg of the dried samples were weighed in sealed aluminum pans for analysis and heated from 30 to 170 °C at a rate of 10 °C using an empty aluminum pan as reference. Nitrogen flow rate of 20 mL/min was used to maintain an inert atmosphere in the sample chamber. DSC Pyris software allowed us to record changes in the heat flow of the samples and data process as thermograms. The physical mixture was prepared by simply blending efavirenz with raw materials (MCT oil, crude soy lecithin, Tween 80 and NaCl-water) at 35 °C, in amounts equivalent to the ones used to form an optimized EFV-LNC formulation.

##### X-ray Diffraction

The crystallinity of EFV-LNC was compared to that of both free efavirenz and blank-LNCs by using X-ray powder diffraction (XRD) on a Bruker D8 Discovery equipped with a Lynx Eye detector (proportional counter), using a nickel filter and Cu-Ka radiation at 1.5404 Å. The scans were run at 2θ range 10–60° with a slit width of 6.0 mm at a scanning speed of 1° min^−1^. Both EFV-LNCs and blank-LNC were freeze-dried prior to the analyses.

##### Fourier Transform Infrared Spectroscopy

The PerkinElmer Spectrum 100 FTIR Spectrometer allowed us to obtain the sample’s IR spectra in attenuated total reflection mode. Sixteen scans were applied in the wavenumber ranging from 650 to 4000 cm^−1^. The signal from functional groups in the freeze-dried EFV-LNC were compared to those of freeze-dried blank-LNC and raw materials (free efavirenz, MCT oil, crude soy lecithin and Tween).

##### Energy-Dispersive X-ray Spectroscopy

The energy-dispersive X-ray spectroscopy was used to perform the surface elemental analysis of EFV-LNCs in comparison to blank-LNC and raw materials (free efavirenz, crude soy lecithin, Tween 80 and NaCl).

#### 2.2.6. In Vitro Release

The optimized formulation with an efavirenz concentration of 17.5 mg/mL was considered for in vitro release studies. The release medium consisted of 1% sodium lauryl sulfate solution. The medium pH was adjusted to 7 then to 4 using concentrated HCl to ascertain the release behavior in both neutral and acidic conditions. Briefly, 200 µL corresponding to 3.5 mg of efavirenz was placed in a dialysis bag (dialysis tubing cellulose membrane, flat width 25 mm (1.0 in), Sigma-Aldrich, St. Louis, MO, USA), which was soaked in 25 mL of release medium. Samples were submerged at 37 °C for 74 h while maintaining a constant shaking (100 rpm). At selected time intervals (0.5, 1, 2, 6, 10, 16, 21, 51 and 74), a 5 mL aliquot of the solution was withdrawn and immediately substituted with 5 mL of fresh release medium to maintain sink conditions. “Factorial multilevel categoric” design was previously performed and generated 66 runs corresponding to a triplicate for each level in both pH 4 and pH 7. The number of levels was 11 corresponding to the time intervals, and the design had two “categoric” factors corresponding to pHs 4 and 7.

#### 2.2.7. Statistical Analysis

Statistical analyses were performed using the two-way analysis of variance (ANOVA), and significance was tested at the 0.05 level of probability. Additionally, certain software packages, such as Design Expert, Origin, and Image-J were, used in data treatment.

## 3. Results and Discussion

### 3.1. Solubility Assessment of Efavirenz

As highlighted in Figure 1, Labrafac Lipophile 1349 displayed the highest solubility for efavirenz (3336 ± 4.27 mg/mL), followed by MCT oil (27,426 ± 7.97 mg/mL). The lowest solubility was observed in HPLC-grade water (0.089 ± 0.0044 mg/mL) confirming the practically insoluble nature of this drug already reported in the literature [25]. Labrafac Lipophile 1349 and MCT oil are triglyceride-based oils. The solubilizing capacity of triglycerides is highly influenced by their intrinsic composition, namely, the degree of unsaturation of their carbon chains. The lower the degree of unsaturation, the higher the solubility [26,27,28]. Both Labrafac Lipophile 1349 and MCT oil are made of saturated fatty acids (capric and caprylic fatty acids). The difference in the solubility of efavirenz in these two oils could be accounted for by their manufacturing process. Labrafac Lipophile is claimed to be originated from strictly vegetal raw materials (not specified) and contains up to 80% and up to 50% of caprylic and capric acid, respectively. MCT oil also originates from vegetal raw material, namely, coconut oil and palm kernel, and is claimed to contain 60% of caprylic and 40% of capric acid. Based on these results and given its availability and affordability on the local market, MCT oil was selected for further investigations.

### 3.2. Statistical Analysis and Optimization of Lipid Nanocapsules

Design-Expert software version 13.0 (Stat-Ease, Inc., Minneapolis, MN, USA) was applied to the design and optimization studies of LNCs. Based on the constraints previously summarized in Table 1 (I-Optimal Mixture Design and Statistical Optimization section), a single-block I-optimal mixture design was launched to ascertain the interactive effects of the different proportions of the mixture components. The I-optimal design provides lower average prediction variance across the region of experimentation and was therefore suitable for this study as the prediction of LNC DS, PDI and ZP is important. The I-optimal design allows best fitting the different responses to statistical models, such as linear, quadratic and special cubic models, for points prediction and optimization [24]. The optimal custom design algorithm generated 24 blends summarized in Table 3, along with their respective values for the droplet size, polydispersity index and zeta potential.

The precision of the design for predictions at different points in the design space was evaluated, and the resulting triangular contour plot as well as the surface plots of the standard error are presented in Figure 2. Given that the errors are almost uniform and relatively small (0.5–1.5 σ, where σ is the estimated variability of the data) across the region of interest, the fitted models are expected to provide precise predictions.

The design software automatically fitted the experimental data to different statistical models, including linear, quadratic, special cubic and cubic models. Statistical parameters, such as the predicted residual sum of squared (PRESS), the lack of fit and the adjusted and predicted R squared were analyzed and some model reductions were applied to obtain best fit as well as best prediction ability. Models with the lowest values for the PRESS and high values for both adjusted and predicted R squared are associated with good prediction ability for a set of data [27]. Table 4 and Table 5, respectively, summarize the ANOVA analysis of the suggested models and that of reduced models that best fit the data. Table 6 highlights the PRESS values of modified models as compared to PRESS values of suggested polynomials.

The droplet size, zeta potential and temperature of dilution were originally fitted to Scheffé special cubic models prior to their reduction for best fit and better prediction ability. The polydispersity index was fitted to a linear model. These fitted models were subject to model *p*-value tests to ascertain the assumption of their prediction ability, and lack of fit F-test to evaluate the variation of data around them. All models were significant (model *p*-value < 0.05), assuming that they could be used to predict data and navigate the design space. Unlike the models fitted for zeta potential and temperature of dilution whose lack of fit was significant (*p*-value < 0.05), an insignificant lack of fit (*p*-value > 0.10) was observed for the reduced special cubic and linear models, respectively, fitted to the droplet size and the polydispersity index, suggesting that the models’ predicted data fit the actual experimental response data [21,29,30]. Prior to the prediction of the optimized formulation, the analysis of residuals was performed for the statistical diagnosis of the models, and the Box–Cox plots for power transformation confirmed that no transformation was required for any of the models.

### 3.3. Droplet Size, Polydispersity Index, Zeta Potential and Temperature of Dilution

Design Expert Software allowed fitting the droplet size, zeta potential and temperature of dilution to Scheffé’s “reduced special cubic polynomials”, while the polydispersity index was fitted to a linear polynomial. These ones are expressed by the following equations:Droplet size = −215.99 × A + 1809.66 × B + 49.4047 × C + −8.08815 × D + −29.3894 × AB + 13.0495 × AC + 3.72536 × AD + −103,505 × BC + −19.1016 × BD + −0.0407471 × CD + 1.68149 × ABC + −0.232067 × ACD + 1.05428 × BCD(3)
Zeta potential = 3159.65 × A + 9637.82 × B + 1599.49 × C + 94.2751 × D + −762,856 × AB + −173,983 × AC + −43.9283 × AD + −126.84 × BC + −112,035 × BD + −23.3279 × CD + 9.63158 × ABC + 8.43236 × ABD + 2.02236 × ACD(4)
Temperature of dilution = 221,779 × A + 45.6503 × B + 261,092 × C + 11.4998 × D + −27,609 × AC + −3.43182 × AD + −1.01391 x BD + −3.9841 × CD + 0.381584 × ACD(5)
Polydispersity index = −0.0137378 × A + 0.000328786 × B + 0.01033 × C + 0.00203285 × D(6)
where factors A, B, C and D stand for MCT oil, crude soy lecithin, Tween 80 and NaCl-water, respectively. These equations in terms of actual factors can be used to make predictions about the response for given levels of each factor. Here, the levels should be specified in the original units (%) for each factor.

As illustrated in Table 7, the experimental droplet size ranged from 29 to 72 nm. The ANOVA analysis revealed that the terms AB, BC, BD (*p*-value < 0.0001) and CD, ABC and BCD (*p*-value < 0.05) significantly influenced the model fitted to the droplet size. This corroborates the fact that changes in the proportions of A, B and C highly affect the droplet size. The droplet-size contour plot is shown in Figure 3a where the dark blue region corresponds to the smallest sizes as the proportions of both Tween 80 and crude soy lecithin increase, while MCT oil proportion is the lowest. Hence, it is interesting to note that the smallest size was observed for the blend where the highest proportion of both crude soy lecithin and Tween 80 were used with the lowest proportion of MCT oil (Table 3). Table 4 shows that the droplet size reduced special cubic model R squared value was 0.9821, meaning that this model could explain 98.21% of the variation in the response, therefore indicating the relevance of the model. The predicted R squared value of 0.876 is in reasonable agreement with the adjusted R squared value of 0.9626 (difference is less than 0.2), implying that the responses’ trends could be analyzed by this model [30,31].

On the other hand, as depicted in Figure 3b, the polydispersity index < 0.2 was observed for all the studied formulations suggesting a good uniformity of size within the particles in the PNCs. The linear model shown in Table 4 is associated with very low adjusted and predicted R squared values, making this model not very strong for predictions. Nevertheless, the adequate precision of 6.7749 indicates an adequate signal; this value must be greater than “4” to explain the signal-to-noise ratio for excellent navigation in the design space [32]. Moreover, the model F-value of 4.69 and not significant lack of fit (*p*-value 0.4584), respectively, imply that the model is significant and that its predicted data fit the actual experimental response data [21,29,30]. However, given that this model could only explain 41.28% of the variation in the response (R^2^ = 0.4128), model validation might help to assess this model’s reliability for polydispersity index predictions [21].

Interestingly, the zeta potential values obtained suggest that the developed LNCs were likely to achieve good stability. As shown in Table 7, the zeta potential ranges from −73 to −35 mV and its fitted model is significantly (*p*-value < 0.05) affected by the terms AB, AC, AD, BC, BD, CD, ABC and ACD. Negatively charged nanoparticles are likely to undergo macrophage uptake and allow drug targeted delivery to action sites [33,34,35]. Macrophages play a pivotal role in viral infections, given that various viruses utilize them as reservoirs to replicate, disseminate, persist or trigger hazardous inflammatory responses [36,37].

As to the data presented in Table 5, the reduced special cubic model could explain 83.93% of the variation in the response (R^2^ = 0.8393). A negative predicted R squared was observed, suggesting that the overall mean may be a better predictor of the zeta potential than the present model.

As far as the temperature of the dilution is concerned, it was found to be significantly affected by crude soy lecithin and NaCl-water (model term BD *p*-value was 0.0126). The terms B and D, respectively, represent crude soy lecithin and NaCl-water. As depicted in Figure 4, the higher the crude soy lecithin proportion, the lower the temperature of the dilution. Dilution was applied at the phase inversion temperature, and phase inversion is observed when microemulsions form, resulting in translucent mixtures. Bi-continuous microemulsion structures are likely to be formed when the non-ionic surfactant is trapped at the interface [14]. Since the higher the amount of non-ionic surfactants, the more likely they are to be trapped at the interface; high amounts of crude soy lecithin are likely to be associated with a low dilution temperature. On the other hand, as the concentration of NaCl increased in the NaCl-water, the temperature of the dilution decreased. This corroborates with the previous studies in which high concentrations of NaCl were associated with a decrease in the phase inversion temperature [13,38].

### 3.4. Model Optimization

In order to evaluate the polynomial models fitted to the droplet size, polydispersity index, zeta potential and temperature of dilution, four blends generated by the design expert software were selected and run in triplicate. MCT oil, lecithin, Tween 80 and NaCl-Water proportions (%) were, respectively, 12:3:9:76 for blend 1, 12:1.5:11.7:74.8 for blend 2, 10:3:13.1:73.9 for blend 3 and 10:1.5:12.9:75.6 for blend 4. As depicted in Table 8, all the average values (replicate = 3) of the experimental data were within the range of the 95% prediction intervals, suggesting a strong prediction ability for the polynomials models fitted to the responses. Given its lowest proportion of ionic surfactant (Tween 80), its ability to produce good particle-size uniformity (PDI = 0.12), as well as its high oil content susceptible to promote drug solubility in the nanocarrier, blend 1 was selected for further investigation, namely, the encapsulation efficiency and drug loading capacity evaluation.

### 3.5. Encapsulation Efficacy and Drug Loading Capacity

#### 3.5.1. Statistical Analysis

In order to determine the amount of efavirenz that LNCs are likely to entrap, their encapsulation efficiency and drug loading capacity were evaluated. The optimized blend 1, from the previous Section 3.4, was used for this purpose. This one consisted of a mixture of MCT oil, crude soy lecithin, Tween 80 and NaCl-water in a ratio of 12:3:9:76% (m/m). Furthermore, in order to build statistical models capable of predicting amounts of drug that would ensure both a high encapsulation efficiency and drug loading capacity in blend 1, the D-optimal (custom) randomized design generated 13 formulation runs detailed in Table 9, together with the experimental data obtained.

The encapsulation efficiency (EE%) and drug loading capacity (DLC%) were automatically fitted to sixth order polynomial models. The different statistical parameters of these models are detailed in Table 10. These were significant (model *p*-value < 0.05) and had predicted data fitting the actual experimental response data (lack-of-fit *p*-value > 0.05). However, they exhibited a low prediction ability as expressed by the high PRESS values along with predicted R² values that are not in reasonable agreement with the adjusted R² values (difference was more than 0.2), implying that response trends could not be analyzed by these models. Hence, model reduction was applied. It consisted of removing terms to improve the adjusted R² value, followed by the removal of terms with *p*-values > 0.100000. As a result, the encapsulation efficiency and the drug loading capacity were, respectively, fitted to a linear model and a quadratic model as expressed by the equations below:EE% = 120.562 + −0.273254 × A and DLC% = −0.390218 + 0.0215689 × A + −5.41283 × 10^−5^ × A^2^(7)
where A stands for the amount of efavirenz.

These reduced models were significant (model *p*-value < 0.05) and had predicted R² values in reasonable agreement with the adjusted R² values (difference was less than 0.2). Adequate precision values higher than four indicated adequate signals, meaning that they could explain the signal-to-noise ratio for excellent navigation in the design space. However, the predicted data of these models could not fit the actual experimental data (lack-of-fit *p*-value < 0.05). Hence, model validation was performed to assess their prediction performance on the encapsulation efficiency and drug loading capacity (see Section 3.5.2).

Interestingly, as illustrated in Figure 5, unlike the drug loading, as the amount of drug increases, the encapsulation efficiency decreases. This can be explained by the solubility of efavirenz in the liquid lipid core of LNCs (see Section 3.1). As the LNC core becomes saturated, no more efavirenz dissolves. This means that dissolved efavirenz is at chemical equilibrium with an excess of undissolved efavirenz [39]. Therefore, the higher the undissolved amount of efavirenz, the more likely it is to be detected in high concentrations in the supernatant when determining the encapsulation efficiency, as previously described.

#### 3.5.2. Model Validation

This section aims to confirm the prediction performance assumption of the models fitted to the encapsulation efficiency and the drug loading capacity. The target was also to obtain an optimized amount of efavirenz that could achieve both high encapsulation efficiency and drug loading capacity in blend 1. Hence, the optimization criteria consisted of maximizing encapsulation efficiency and drug loading capacity values with the amount of efavirenz minimized to avoid excess undissolved drug. The software generated only one amount of drug that was formulated in blend 1 in triplicate. The formulation was performed using the method described in Section 2.2.4, by adding 135 mg of efavirenz to blend 1. Table 11 shows that the average value of the experimental data was within the range of the 95% prediction intervals, suggesting that the models fitted to the responses could be used to predict the encapsulation efficiency as well as the drug loading capacity. This amount of drug was selected for drug-release investigation in blend 1.

### 3.6. Characterization of Blank-LNCs and EFV-LNCs

#### 3.6.1. Droplet Size and Shape Analysis

Both blank-LNC and EFV-LNC were characterized by very small droplets of various sizes, as demonstrated by DLS (dynamic light scattering) Gaussian distribution in Figure 6A. Moreover, the average size of blank-LNC was around 42 nm, which increased to over 80 nm following EFV encapsulation. This was confirmed by TEM analysis that describes, as depicted in Figure 6B, normal distribution peaks at around 30 to 50 nm (Figure 6C) and 50 to 80 nm (Figure 6E) for blank-LNC and EFV-LNC, respectively. Moreover, as illustrated in Figure 6B,D, different particle shapes were observed from the TEM images, the majority of the particles being nearly ellipsoidal and spherical, as already observed in the previous studies [13,40]. It is noteworthy to mention that EFV encapsulation resulted in the growth of particle shapes, as shown in the TEM images.

#### 3.6.2. Diffraction Scanning Calorimetry

DSC analysis was performed in order to investigate any possible drug-excipient interaction, as well as any change in the crystalline nature of the drug as LNCs form. Figure 7 shows the overlaid thermograms of efavirenz, physical mixture of LNC components, blank-LNC and EFV-LNC. At first sight, an endothermic peak at approximately 138 °C, associated with 48.875 J/g enthalpy, indicated the melting point of efavirenz, in agreement with the previous studies [41,42,43]. The observed sharp peak suggests high drug purity and crystallinity. On the other hand, the shift in the appearance of this peak from 138.67 °C to 122.33 °C in the physical mixture thermogram suggests possible interactions that could be a result of hydrogen bond formations between the drug and the raw materials. However, it could also be a result of the amorphous form formation as EFV quickly dissolves in the physical mixture, thus accounting for a low melting energy consumption (ΔH = 6.8802 J/g). Such outcomes have been reported when EFV was formulated in polyvibylpyrrolidone or in a binary mixture with nicotinamide [41,42]. Interestingly, no major peak has been observed either for blank-LNC or for EFV-LNC over the scanned temperature range, suggesting an excellent encapsulation of efavirenz in an amorphous or molecular dispersed nature with the advantage of enhancing its solubility and dissolution in biological media.

#### 3.6.3. Fourier Transform Infrared Spectroscopy

The structure of EFV includes a secondary amine that may act as a weak Bronsted–Lowry acid and a carbonyl group (C=O), as well as a C–O–C group, which are acceptor groups. In addition, MCT oil, Tween 80 and soy lecithin have in common carbonyl groups as acceptors of hydrogen bonding, which could favor some interactions with EFV [44]. These interactions may be expressed by the shifting or broadening of bands, disappearance of peaks or intensity alterations [45]. Hence, the FTIR spectra were recorded to assess and scout possible chemical interactions between the pure drug and raw materials in the developed nanocapsules.

The different spectra obtained are shown in Figure 8. As can be observed from the graphs, LNC formulations exhibit intense characteristic bands of aliphatic groups, which are a clear contribution of the structural composition of soy lecithin, Tween 80 and MCT oil. The C–H stretch of saturated fatty acids was observed at around 2950–2840 cm^−1^. Bands at 1750–1745 cm^−1^ and 1416 cm^−1^, respectively, revealed the presence of carbonyl functions (C=O) and C–O–C stretching vibrations, which are both part of the structure of EFV and the raw materials. The broad band visible in the range of 3700–3000 cm^−1^ is associated with O–H stretching in the spectra of Tween 80 and soy lecithin. On the other hand, all the expected signals for pure EFV were present. They encompass the N–H stretching vibrations and N–H bending vibrations of benzoxazin-2-one ring, respectively, at about 3320 cm^−1^ and 1600 cm^−1^; the exocyclic tricyclic triple bond (–C≡C–) at around 2250 cm^−1^; and C–F and C–Cl stretching near 1250 cm^−1^ and 1038, respectively, in agreement with the previous studies [45,46,47,48].

As clearly displayed in Figure 8, the disappearance of N–H stretching and bending vibrations, as well as a plunge in the intensity of carbonyl group stretches, occurred in the EFV-based LNCs and the physical mixture. These events could result from the physical interactions between EFV and the raw materials, suggesting intramolecular hydrogen bonding. They could also be a result of the transition from the crystalline form of EFV to an amorphous form, as confirmed by the XRD and DSC data. Moreover, the O–H stretch of the terminal hydroxylic group of Tween 80 and soy lecithin vanished in the developed nanocapsules and the physical mixture, thus indicating intramolecular hydrogen bonding between the drug and these two raw materials. The absence of the characteristic peak of the exocyclic tricyclic triple bond (–C≡C–) that appeared at around 2250 cm^−1^ indicates the successful entrapment of EFV in the lipid core of LNCs.

#### 3.6.4. X-ray Diffraction

Changes in the crystalline structure of EFV in lipid nanocapsules was ascertained by using powder X-ray diffraction. Overall, the LNC formulation influenced EFV crystallinity. The diffractograms sketched in Figure 9 unveil a very crystalline state (88.1%) of efavirenz (EFV) characterized by remarkable peaks in 2 tetha diffraction angle ranges of almost 8 to 25°, in accordance with the previous investigations [41,45]. The blank-LNC formulation exhibited few diffraction peaks with a relatively low crystallinity of 23.7%, which increased to 52.3% following EFV encapsulation. The disappearance of EFV characteristic diffraction peaks in the EFV-LNC formulation indicates that EFV was successfully entrapped in the LNC matrix. This result corroborates with the previous DSC conclusions and further supports the EFV transformation into its amorphous or molecular dispersed nature, hence accounting for the enhancement of its solubility.

#### 3.6.5. Energy-Dispersive X-ray Spectroscopy

Energy-dispersive X-ray spectroscopy was used to probe the qualitative composition of freeze-dried EFV-LNC and freeze-dried blank-LNC, as compared to that of free efavirenz. The EDS spectra of both blank-LNC and EFV-LNC were characterized in common by the presence of carbon, chlorine, oxygen, sodium and phosphorus (Figure 10b,c). Fluorine was observed only in EFV-LNC as being characteristic of the structure of efavirenz (Figure 10a,c). In particular, phosphorus characterized the LNC formulations as being a key element in the hydrophilic heads of phospholipids that are organized along with the hydrophilic heads of the ionic surfactant (Tween 80) to form the rigid shell of LNCs [13,49].

### 3.7. In Vitro Release

The choice of a model drug is very critical in the design of a drug delivery system, given that its stability and activity must be maintained following encapsulation. Many factors, especially the components used to design the carrier, could interact with the drug and negatively affect the stability of its structure as well as its biological activity [50]. To verify whether the structure of efavirenz remained intact following encapsulation in LNCs, the release medium containing the liberated drug was collected and the drug spectrum was recorded using a UV spectrometer. As displayed in Figure 11, no change was observed in the efavirenz spectrum (λmax = 247 nm) following drug release, as compared to its spectrum before drug release, hence suggesting that the structural integrity of the drug was preserved during the entire EFV-LNC manufacturing process.

The in vitro release profiles of EFV-LNC and free EFV are sketched in Figure 12. Overall, the release profile in pH 7 was higher than in pH 4 for EFV-LNC. As illustrated by the interactive graphs below (Figure 12a), free EFV was quickly released from the dialysis bags, as compared to EFV-LNC both in pH 4 and 7, reaching up to around 40% release in 74 h. This could be explained by the fact that, before crossing the dialysis bag barrier, encapsulated EFV had to be first liberated from the LNC carrier.

On the other hand, EFV-LNC liberated the drug quicker in pH 7 than in pH 4, as shown by the interactive plot (Figure 12a). However, both release profiles in pH 4 and 7 followed an upward trend up to the end of the experiment, suggesting a prolonged release that could be complete if more time was set up. From a statistical point of view, the analysis of variance revealed that both the release time and pH of the release medium significantly affected EFV liberation from the LNCs (*p*-value < 0.05). Finally, as outlined in the normal plot of residuals (Figure 13a for EFV-LNC and Figure 13b for free EFV), a linear regression was observed, therefore supporting the fact that the residuals were normally distributed.

## 4. Stability Studies

The developed LNCs are not true solutions and are, rather, a homogenous colloidal system consisting of a dispersion of different substances; hence, they are likely to lose their stability over time. Particles in a colloidal system are influenced by diverse types of forces that could either foster their stability or cause their flocculation. According to the Derjaguin, Landau, Verwey and Overbeek theory, these forces include the Van Der Walls attraction forces and the repulsive electric double-layer forces [51]. Briefly, if the electrostatic repulsive forces are higher than the attractive ones, no flocculation is expected to occur and the system can be considered as stable. This phenomenon is known as electrostatic stabilization [52]. In other words, when particles in suspension have a similar electrostatic charge on the surface, the system stability increases as no attraction occurs between them. As depicted in the bar charts in Figure 14, the EFV-LNC remains stable over 28 days of stability studies as referring to the zeta potential measurement values (Figure 14c), knowing that zeta potential represents the electric surface properties of particles in a colloidal system [53,54]. On the other hand, the droplet size remained stable (Figure 14a) and no change in the polydispersity index was observed over the period of experimentation (Figure 14b).

## 5. Conclusions

The present work reported the successful encapsulation of EFV in lipid nanocapsules and its prolonged release from this matrix. The design expert software allowed the building of polynomial models that could predict the different characteristics of these nanomaterials, namely, the particle size, polydispersity index and zeta potential. Different characterization techniques confirmed the entrapment of EFV in the lipid nanocapsules and its transformation into its amorphous state, therefore enhancing its solubility. Among them, FTIR revealed some chemical interactions, assumed as hydrogen bonds, between the drug and the different starting materials owing to their respective structures. Stability studies revealed that the developed LNC were quite stable over the period of experimentation. Overall, lipid nanocapsules stand as promising delivery system for the improvement of EFV biopharmaceutical properties. Further investigations are ongoing in our laboratories to evaluate the antiviral activities of the formulated nanocapsules as compared to that of the free EFV.

## Figures and Tables

**Figure 1 pharmaceutics-14-01318-f001:**
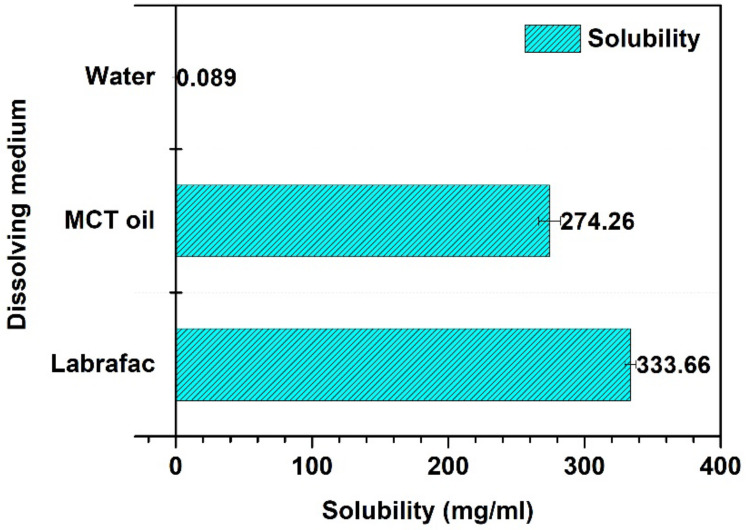
Solubility profile of efavirenz in Labrafac Lipophile 1349, MCT oil and HPLC-grade water.

**Figure 2 pharmaceutics-14-01318-f002:**
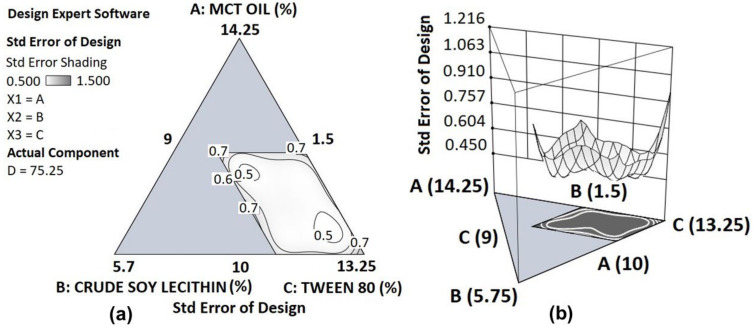
Contour plot (**a**) as well as the surface plot (**b**) of the standard error at different points in the design space with changing compositions of MCT oil (A), crude soy lecithin (B) and Tween 80 (C), and with fixed composition of NaCl-water (D).

**Figure 3 pharmaceutics-14-01318-f003:**
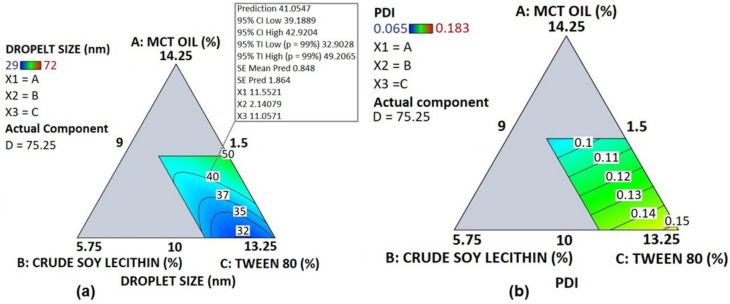
Contour plot of droplet size (**a**) and polydispersity index (**b**).

**Figure 4 pharmaceutics-14-01318-f004:**
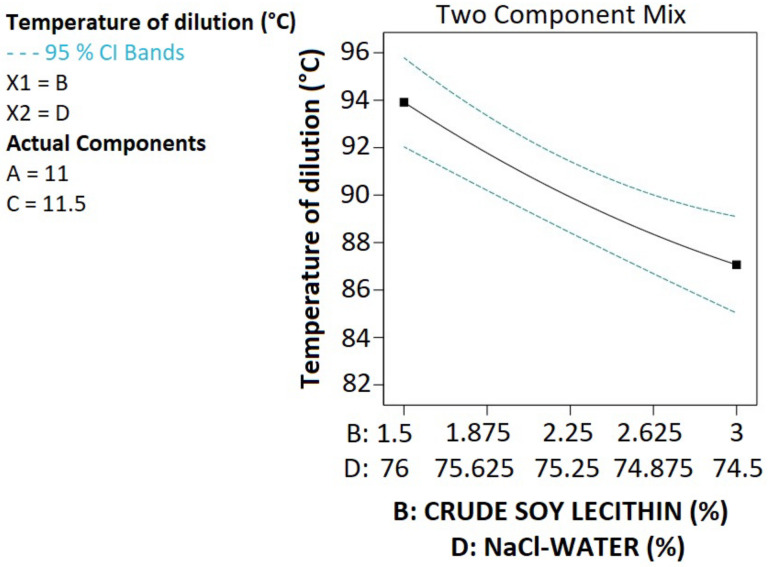
Impact of crude soy lecithin and NaCl-water on temperature of dilution with MCT oil (A) and Tween 80 (C) proportions set, fixed at 11 and 11.5%, respectively.

**Figure 5 pharmaceutics-14-01318-f005:**
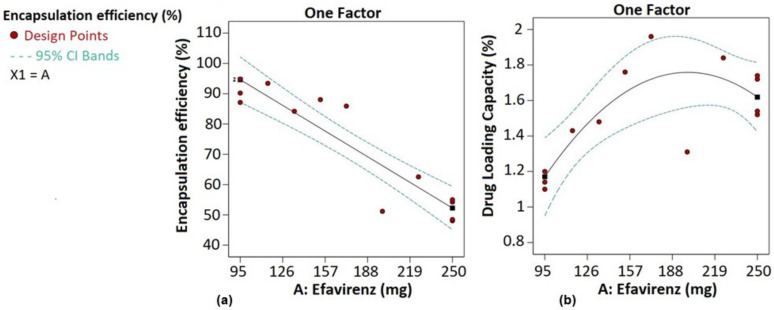
Encapsulation efficiency (**a**) and drug loading capacity (**b**) profiles.

**Figure 6 pharmaceutics-14-01318-f006:**
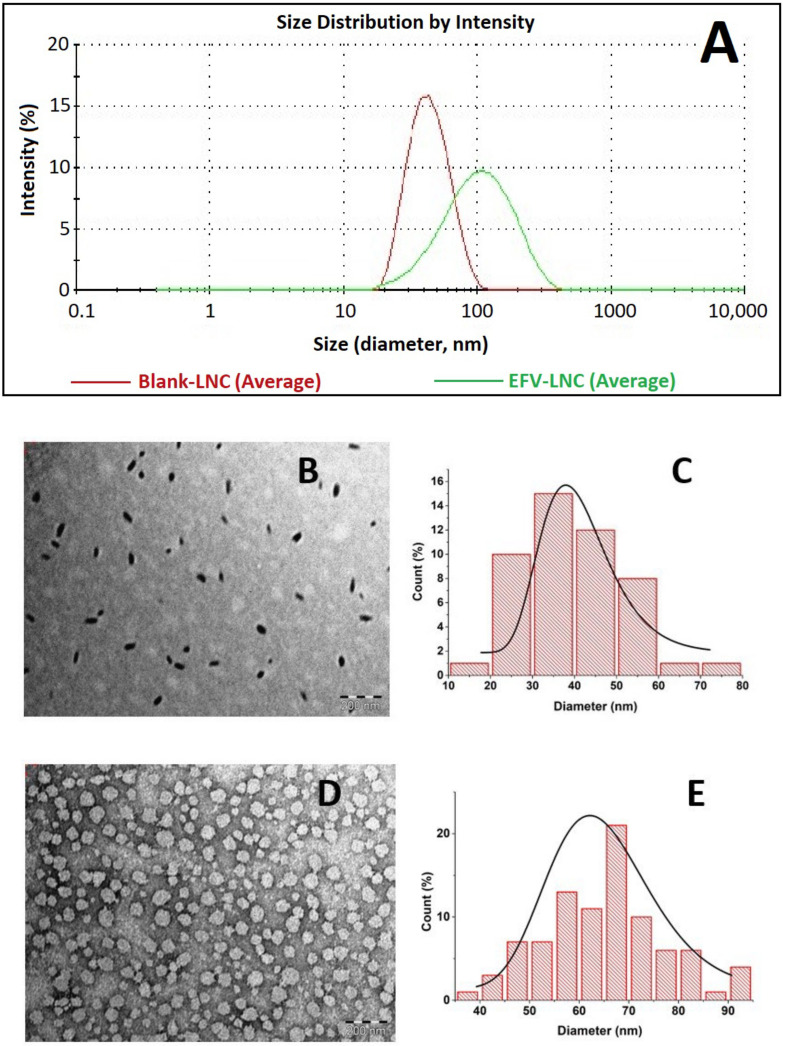
DLS average-particle-size distribution (triplicate) of blank-LNC and EFV-LNC (**A**), TEM images (200 nm scale) of blank-LNC (**B**) and EFV-LNC (**D**), TEM particle-size distribution by ImageJ of blank-LNC (**C**) and EFV-LNC (**E**).

**Figure 7 pharmaceutics-14-01318-f007:**
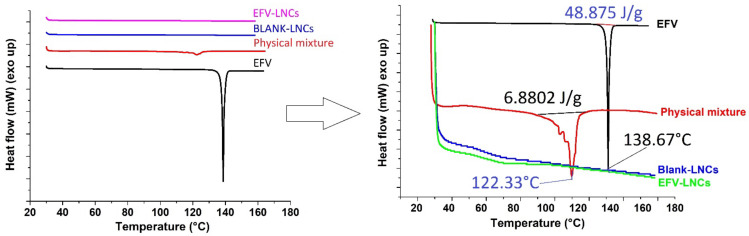
DSC thermograms.

**Figure 8 pharmaceutics-14-01318-f008:**
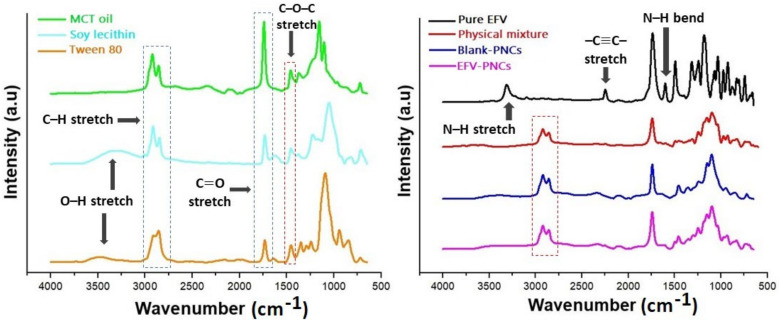
FTIR spectra of LNC formulations and raw materials.

**Figure 9 pharmaceutics-14-01318-f009:**
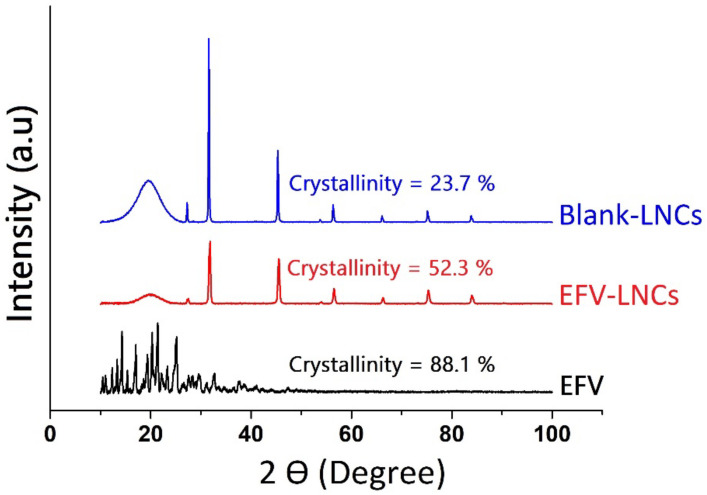
XRD diffractograms of blank-LNC, EFV-LNCs and EFV.

**Figure 10 pharmaceutics-14-01318-f010:**
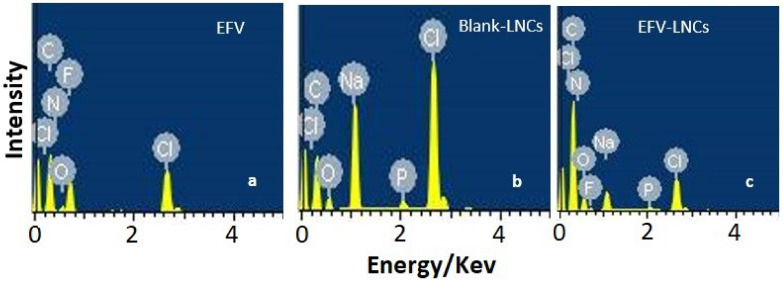
EDS spectra illustrating the elemental composition of free efavirenz (**a**) and LNC formulations (**b**,**c**).

**Figure 11 pharmaceutics-14-01318-f011:**
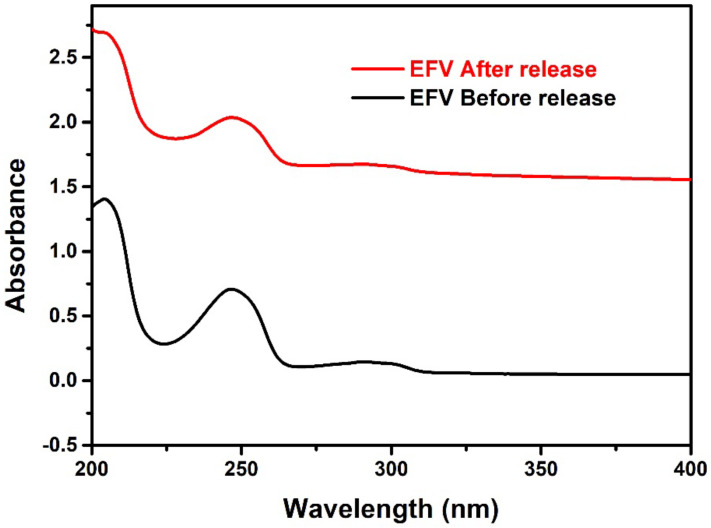
Efavirenz UV spectra before encapsulation and after release from EFV-LNC.

**Figure 12 pharmaceutics-14-01318-f012:**
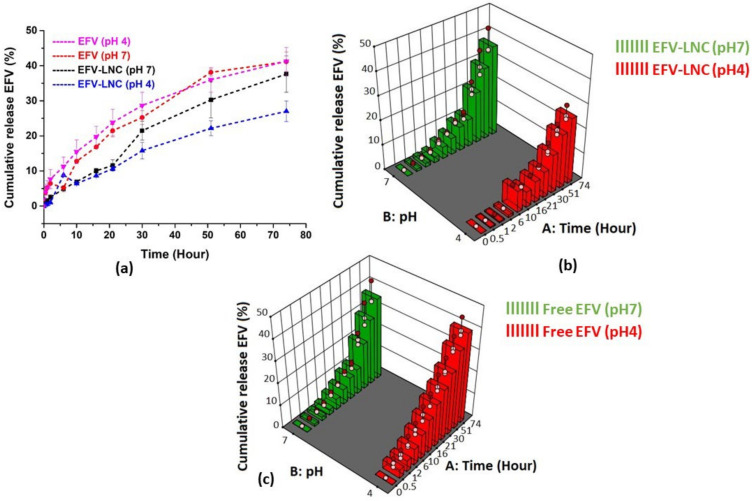
Interactive plot (**a**) and 3D surface plots of EFV-LNC (**b**) and free EFV (**c**) in vitro release profiles in both pH 7 and pH 4.

**Figure 13 pharmaceutics-14-01318-f013:**
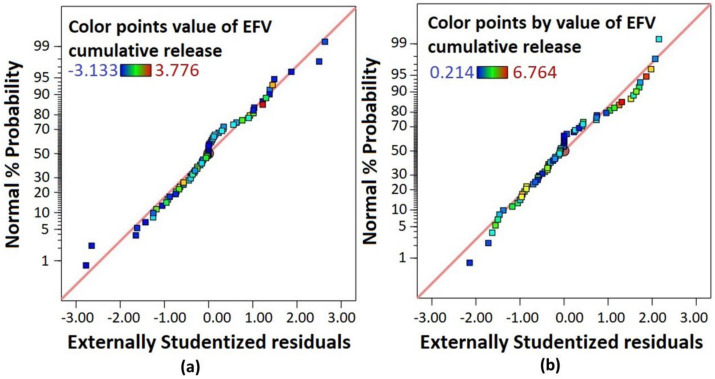
Normal plot of residuals for EFV-LNC (**a**) and for free EFV (**b**).

**Figure 14 pharmaceutics-14-01318-f014:**
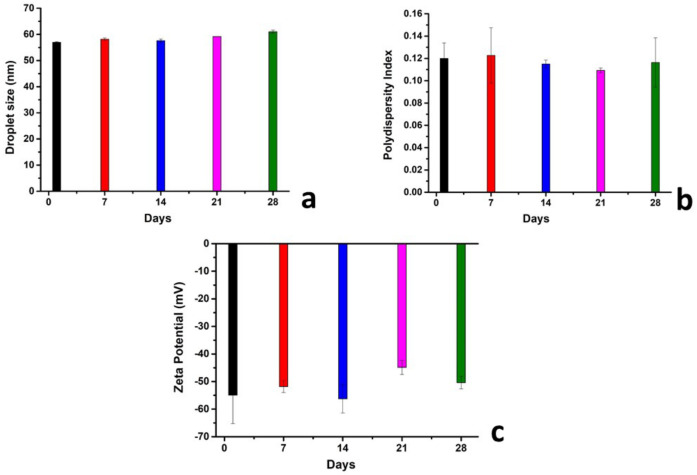
Droplet size (**a**), polydispersity index (**b**), zeta potential (**c**) values over 28 days of EFV-LNC stability evaluation.

**Table 1 pharmaceutics-14-01318-t001:** Input variable constraints for I-optimal mixture design.

Low Limit		Constraint		High Limit
10,000	≤	A: PURE MCT OIL	≤	12,000
1500	≤	B: CRUDE SOY LECITHIN	≤	3000
9000	≤	C: TWEEN 80	≤	14,000
71,000	≤	D: NaCl-WATER	≤	79,500
		A + B + C + D	=	100,000

**Table 2 pharmaceutics-14-01318-t002:** Optimization criteria and generated blends.

Components (%)	Blend 1	Blend 2	Blend 3	Blend 4
Criteria	A maximized and C minimized		A minimized and C maximized	
Unit	%		%	
MCT oil (A)	12.00	12.00	10.00	10.00
Crude soy lecithin (B)	3.00	1.50	3.00	1.50
Tween 80 (C)	9.00	11.70	13.10	12.94
NaCl-water (D)	76.00	74.80	73.90	75.56

**Table 3 pharmaceutics-14-01318-t003:** Blend compositions generated by I-optimal design and observed experimental data.

Input Variables (% m/m)					Responses			
	**MCT Oil** **A**	**Crude Soy Lecithin** **B**	**Tween 80** **C**	**Salt Water** **D**	**Droplet Size** **(nm)**	**PDI**	**Zeta Potential** **(mV)**	**Temperature of Dilution ** **(°C)**
1	12.00	2.16	11.50	74.34	40	0.093	−49	93
2	10.00	2.42	11.49	76.09	33	0.127	−73	83
3	11.16	1.50	11.38	75.95	46	0.152	−56	95
4	11.83	2.87	13.89	71.41	31	0110	−50	86
5	10.00	3.00	9.00	78.00	34	0.097	−37	85
6	10.68	1.99	14.00	73.33	33	0.128	−46	87
7	10.00	1.50	13.07	75.44	35	0.161	−48	90
8	10.00	3.00	14.00	73.00	29	0.183	−56	85
9	11.16	1.50	11.38	75.95	46	0.149	−57	95
10	12.00	1.50	9.00	77.50	72	0.065	−54	95
11	10.21	1.64	9.03	79.13	46	0.090	−35	95
12	12.00	2.16	11.50	74.34	40	0.136	−55	91
13	12.00	3.00	9.55	75.45	42	0.073	−53	89
14	11.79	1.50	12.92	73.79	43	0.101	−39	92
15	10.00	2.96	12.83	74.22	35	0.137	−56	86
16	10.97	3.00	11.45	74.58	39	0.113	−49	86
17	11.27	2.42	9.00	77.31	41	0.147	−64	87
18	10.25	1.50	10.53	77.72	40	0.089	−36	93
19	12.00	1.50	14.00	72.50	42	0.129	−41	89
20	10.67	3.00	10.10	76.22	37	0.099	−52	89
21	10.68	1.99	14.00	73.33	35	0.141	−44	87
22	10.00	2.42	11.49	76.09	33	0.177	−70	85
23	12.00	3.00	12.32	72.68	35	0.108	−52	85
24	11.27	2.42	9.00	77.31	47	0.113	−68	87

**Table 4 pharmaceutics-14-01318-t004:** Analysis of variance of the suggested models.

Response	Suggested Model	f-Value	Degrees of Freedom	*p*-Value	R^2^	Adjusted R^2^	Predicted R^2^	Adequate Precision
Droplet size (nm)	Special cubic	45.80	13	<0.0001	0.9835	0.9620	0.8718	33.2536
PDI	Linear	4.69	3	0.0123	0.4128	0.3247	0.1771	6.7749
Zeta potential (mV)	Special cubic	5.09	13	0.0072	0.687	0.6980	−2.3389	8.9294
Temperature of dilution (°C)	Linear	7.22	13	0.0018	0.9037	0.7785	−0.1615	7.8467

**Table 5 pharmaceutics-14-01318-t005:** Analysis of variance of the modified models.

Response	Reduced Model	f-Value	Degrees of Freedom	*p*-Value	R^2^	Adjusted R^2^	Predicted R^2^	Adequate Precision
Droplet size (nm)	Reduced special cubic	50.28	12	<0.0001	0.9821	0.9626	0.8876	34.5526
PDI	Linear	NM	NM	NM	NM	NM	NM	NM
Zeta potential (mV)	Reduced special cubic	4.79	12	0.0072	0.8393	0.6641	−1.0050	8.3624
Temperature of dilution (°C)	Reduced special cubic	10.79	8	<0.0001	0.8520	0.7730	0.6025	9.3516

NM: Not modified.

**Table 6 pharmaceutics-14-01318-t006:** PRESS values of suggested and modified models.

PRESS Values		
	Suggested Models	Modified Models
Droplet size (nm)	217.02	190.28
Polydispersity index	0.0182	NM
Zeta potential (mV)	8171.50	4906.83
Temperature of dilution (°C)	386.72	132.34

NM: Not modified.

**Table 7 pharmaceutics-14-01318-t007:** Summary of the responses.

Response	Name	Units	Observations	Minimum	Maximum	Mean	Std. Dev.	Ratio
R1	Droplet size	nm	24.00	29	72	39.75	8.58	2.48
R2	PDI		24.00	0.065	0.183	0.1216	0.031	2.82
R3	Zeta potential	mV	24.00	−73	−35	−51.67	10.32	2.09
R4	Temperature of dilution	°C	24.00	83	95	88.96	3.80	1.14

**Table 8 pharmaceutics-14-01318-t008:** Observed means against the prediction intervals.

	Response	Mean		95% Prediction	
		Predicted	Observed	95% PI Low	95% PI High
Blend 1					
MCT oil: 12%	Droplet size (nm)	41.5892	41.3333	37.0875	46.0908
Lecithin: 3%	PDI	0.0835994	0.12	0.0426535	0.124545
Tween 80: 9%	Zeta potential (mV)	−59.8024	−58.6333	−76.1047	−43.5001
NaCl-Water: 71%	Temperature of dilution (°C)	86.272	86	82.2926	90.2514
Blend 2					
MCT oil: 12%	Droplet size (nm)	48.3797	49.7633	44.9551	51.8043
Lecithin: 1.5%	PDI	0.108375	0.141	0.0710762	0.145674
Tween 80: 11.7%	Zeta potential (mV)	−41.9609	−53.5333	−53.8046	−30.1172
NaCl-Water: 74.8%	Temperature of dilution (°C)	94.9996	95	91.9209	98.0782
Blend 3					
MCT oil: 10%	Droplet size (nm)	33.5363	30.52	30.3943	36.6782
Lecithin: 3%	PDI	0.149043	0.179667	0.110426	0.187661
Tween 80: 13.1%	Zeta potential (mV)	−58.7789	−51.8667	−70.0912	−47.4666
NaCl-Water: 73.9%	Temperature of dilution (°C)	84.3132	84	81.3159	87.3105
Blend 4					
MCT oil: 10%	Droplet size (nm)	35.1093	38.5033	31.3772	38.8414
Lecithin: 1.5%	PDI	0.150377	0.177667	0.111305	0.189449
Tween 80: 12.9%	Zeta potential (mV)	−42.8831	−41.2667	−57.0207	−28.7456
NaCl-Water: 75.6%	Temperature of dilution (°C)	88.6115	88	84.9719	92.2511

**Table 9 pharmaceutics-14-01318-t009:** Encapsulation efficacy and drug loading capacity of blend 1.

Input Variables (% m/m)						Response (%)	
	MCT Oil A	Crude Soy Lecithin B	Tween 80 C	Salted Water D	Efavirenz	Encapsulation Efficiency	Drug Loading Capacity
1	12.00	3.00	9.00	76.00	115.15	93.4	1.43
2	12.00	3.00	9.00	76.09	95	90.19	1.14
3	12.00	3.00	9.00	75.95	134.525	84.19	1.48
4	12.00	3.00	9.00	71.41	250	48.44	1.54
5	12.00	3.00	9.00	78.00	172.5	85.89	1.96
6	12.00	3.00	9.00	73.33	198.85	51.15	1.31
7	12.00	3.00	9.00	75.44	153.545	88.03	1.76
8	12.00	3.00	9.00	73.00	250	48.05	1.52
9	12.00	3.00	9.00	75.95	250	54.97	1.74
10	12.00	3.00	9.00	77.50	95	87.1	1.1
11	12.00	3.00	9.00	79.13	95	94.8	1.2
12	12.00	3.00	9.00	74.34	225.2	62.54	1.84
13	12.00	3.00	9.00	75.45	250	54.23	1.72

**Table 10 pharmaceutics-14-01318-t010:** Statistical parameters of polynomial models fitted to encapsulation efficiency and drug loading capacity of blend 1.

	Encapsulation Efficiency		Drug Loading Capacity	
	Sixth Order Model	Linear Model	Sixth Order Model	Quadratic Model
PRESS values	2408.35	789.70	4.43	0.5820
R squared	0.9826	0.8584	0.9393	0.6311
Adjusted R squared	0.9653	0.8455	0.8783	0.5573
Predicted R squared	0.4514	0.8201	−3.8134	0.3669
Adeq Precision	16.1970	14.3630	10.7785	6.6481

**Table 11 pharmaceutics-14-01318-t011:** Optimized encapsulation efficiency and drug loading capacity.

	Response	Mean		95% Prediction	
		Predicted	Observed	95% PI Low	95% PI High
Blend 1-EFV 135	EE%	83.7808	87.4	72.7572	94.8044
	DLC%	1.53233	1.5	1.2386	1.82605

## Data Availability

The data presented in this study are available on request from the corresponding authors.

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
