# Peer review of "Design, Manufacturing, Characterization and Evaluation of Lipid Nanocapsules to Enhance the Biopharmaceutical Properties of Efavirenz"

_pharmaceutics, 2022, doi:10.3390/pharmaceutics14071318_

Round 1

Reviewer 1 Report

The manuscript describes the formulation of lipid nanocapsules for efavirenz. The authors need to address the following comments before the manuscript can be accepted for publication:

1- Page 2 in the introduction section, there is no need to explain the origin of the response surface methodology. 

2- The paper needs extensive language editing, some of the sentences are incomprehensible, and there are lots of spelling mistakes.

3- Freeze drying as a process is known to affect the crystallinity of formulations, since it causes amorphization of samples. How did the authors rely on the results of the XRAY diffraction?

4- The R2 of the PDI and zeta potential models are very low, hence they are not reliable.

5- For the models of zeta potential and temperature, the predicted R2 is negative, what is the meaning of the negative sign?

6-  The TEM picture of the blank LNC is very strange, no spherical particles are evident, in contrast to drug loaded LNC.

7- The authors didn't conduct any in vitro or in vivo experiments to verify that the therapeutic efficacy of the drug was enhanced or even preserved upon loading in LNC. 

Reviewer 2 Report

The manuscript entitled “Lipid-nanocapsules to enhance the biopharmaceutical properties of efavirenz” is very well organized and presents interesting and promising results. Please consider reviewing based on the suggestions below, in order to improve the article.

1. Introduction:

- Please enter bibliographic notes regarding to the advantages of the lipid-nanocapsules (LNC). Also, the introduction section can be completed with several references concerning the role of the chosen drug (Efavirenz-EFV).

2. Materials and Methods

-On page 5 lines 202-203, the authors said that: “The same parameters were evaluated over a period of 28 days for stability studies using samples stored at 15°C+0.8.” Why did they use this temperature? The samples can normally be stored at room temperature or in the refrigerator. Please explain!

3. Results and Discussion

-On section “3.6.1. Droplet size and shape analysis” page 16, the authors mention that: “Both Blank-LNC and EFV-LNC were characterized by very small droplets of uniform size as demonstrated by DLS (Dynamic light scattering) Gaussian distribution in Figure 6A.” If we look carefully at the granulometric distribution curves, especially in the case of EFV-LNC sample, we notice that the allure of the curve is very wide which means that the diameter is not homogeneous as the authors mentioned. Figures 6D and 6B clearly show an inhomogeneous size distribution both in the case of Blank-LNC and in the case of EFV-LNC. Also, LNC have an irregular contour and some are even ellipsoidal in shape, not spherical.

Please review the comment!

- The scale in Figure 10 looks unaesthetic and is incomplete. Please modify!

- The section 3.6. In vitro release must be 3.7. Please renumber!

- Regarding to the “In vitro release” studies, no conclusion can be drawn without the release test for the simple EFV at an equivalent dose to that used in the EFV-LNC sample. Please perform the release of the simple EFV keeping the same conditions!

Round 2

Reviewer 1 Report

The authors have answered my queries.